# Creation of novel alleles of fragrance gene *OsBADH2* in rice through CRISPR/Cas9 mediated gene editing

**Shanthinie Ashokkumar, Deepa Jaganathan, Valarmathi Ramanathan, Hifzur Rahman¤, Rakshana Palaniswamy, Rohit Kambale, Raveendran Muthurajan** *

Department of Plant Biotechnology, Centre for Plant Molecular Biology and Biotechnology, Tamil Nadu Agricultural University, Coimbatore, Tamil Nadu, India

¤ Current address: International Center for Biosaline Agriculture, Dubai, UAE
* raveendrantnau@gmail.com

**Data Availability Statement:** All relevant data are within the paper and its Supporting Information files.

## Abstract

Fragrance in rice grains is a key quality trait determining its acceptability and marketability. Intensive research on rice aroma identified mutations in *betaine aldehyde dehydrogenase* (*OsBADH2*) leading to production of aroma in rice. Gene editing technologies like CRISPR/Cas9 system has opened new avenues for accelerated improvement of rice grain quality through targeted mutagenesis. In this study, we have employed CRISPR/Cas9 tool to create novel alleles of *OsBADH2* leading to introduction of aroma into an elite non-aromatic rice variety ASD16. PCR analysis of putative transformants using primers targeting the flanking regions of sgRNA in the 7th exon of *OsBADH2* identified 37.5% potential multi-allelic mutations in $T_0$ generation. Sensory evaluation test in the leaves of $T_0$ lines identified thirteen lines belonging to five independent events producing aroma. Sequence analysis of these aromatic $T_0$ lines identified 22 different types of mutations located within -17 bp to +15bp of sgRNA region. The -1/-2 bp deletion in the line # 8–19 and -8/-5 bp deletion in the line # 2–16 produced strong aroma and the phenotype was stably inherited in the $T_1$ generation. Comparative volatile profiling detected novel aromatic compounds viz., pyrrolidine, pyridine, pyrazine, pyradazine and pyrozole in the grains of $T_1$ progenies of line # 8–19. This study has demonstrated the use of CRISPR/Cas9 in creating novel alleles of *OsBADH2* to introduce aroma into any non-aromatic rice varieties.

## Introduction

Rice is a staple food for more than half of the world's population [1]. Apart from yield, grain quality traits such as milling %, appearance, grain size, cooking quality and aroma determine the acceptance of a variety by the consumers which in turn determines the adoption of the variety by the farmers and marketability. Aroma is considered as one of the most preferred quality parameters next to cooking quality, taste and elongation after cooking [2]. Fragrant rice varieties are becoming popular not only among the consumers of Asia but also in Europe and USA [3]. More than 200 volatile compounds have been reported to be associated with aroma in rice grains,

**Funding:** RM received a grant (BT/PR25820/GET/ 119/100/2017) from the Department of Biotechnology, Government of India, New Delhi (www.dbtindia.gov.in). The funder had no role in the study design, data collection and analysis, decision to publish and manuscript preparation.

**Competing interests:** The authors have no competing interests

among which 2-acetyl-1-pyrroline (2AP) was reported to be the principal compound producing aroma [4]. In addition to 2AP, several volatiles belonging to classes of hydrocarbons, aldehydes, ketones, esters, alcohols, phenols etc., have also been reported to contribute for rice aroma [5].

It was thought that rice aroma is determined by a single dominant gene [6]. Later, it was reported that aroma in rice grains is controlled by a single recessive gene [7]. Several other studies have reported that aroma in rice grains is a polygenic trait [8–17]. Amarawathi *et al.* [17] mapped three different loci (one each on chromosomes 3, 4 and 8) by utilizing a mapping population derived between Pusa 1121 and Pusa 1342. Among the three loci, *ARO8.1* was found to be consistent across several genetic backgrounds [9, 10]. Further efforts on fine mapping of ARO*8.1* have led to the identification of *BADH2* (betaine aldehyde dehydrogenase) responsible for the production of aroma in rice grains [17, 18]. It is reported that dominant *BADH2* converts γ-aminobutyraldehyde (GABald) to gamma aminobutyric acid (GABA). In the absence of a functional *BADH2* i.e., non-functional *BADH2* with an 8 bp deletion in exon 7 of *BADH2* gene, GABald is converted into an aromatic compound 2-Acetyl 1-Pyrroline [13]. Intensive research on the haplotype diversity of *BADH2* led to the development of user-friendly Indel marker discriminating aromatic and non-aromatic rice genotypes [17] which enabled accelerated development of aromatic rice genotypes through MAS programs.

Development of aromatic rice genotypes possessing high yield and desirable grain quality traits through conventional breeding or marker assisted breeding is time consuming and labor intensive. Few attempts have been made through RNAi technology to reduce the expression levels of *OsBADH2* and thereby increasing the level of 2AP in rice grains [19–22]. However, due to the incomplete suppression of the expression of *OsBADH2* and also due to the regulatory issues, transgenic approach is no longer a preferred technique for aroma improvement in rice. Recent advancements in the genome editing technology using sequence-specific nucleases (SSNs) have paved way for accelerated improvement of complex traits through targeted gene editing. Among the SSNs, Clustered Regularly Interspaced Short Palindromic Repeats (CRISPR)-associated endonuclease Cas9 (CRISPR/Cas9) is becoming popular due to its efficiency in creating precise mutations. CRISPR/Cas9 is increasingly applied in various crop improvement programs including rice [23, 24], maize [25] and wheat [26], tomato [27], soybean [28], citrus [29], cotton [30], and cassava [31] etc.

In rice, several genes such as phytoene desaturase (*OsPDS*), mitogen-activated protein kinase (*OsMPK2*), bacterial blight susceptible genes/sugar transporters (*SWEETs*), Myb family transcription factor (*OsMYB1*), stress-responsive rice mitogen-activated protein kinase (*OsMPK5*), chlorophyll A oxygenase-1 (*CAO1*) and LA1/rice tiller angle controlling gene (*LAZY01*) have been successfully edited using CRISPR/Cas9 system [23, 32, 33]. Shan *et al.* [34] successfully edited *OsBADH2* using transcription activator like effector nucleases (TALENs) and the edited plants were found to possess increased accumulation of 2AP. Due to its technical tediousness, TALEN technology is not preferred for gene editing in plants. Shao *et al.* [35] successfully edited *OsBADH2* in non-aromatic rice Zhonghua 11 using CRISPR/Cas9 which led to increased accumulation of 2AP. The present study was formulated with an aim of creating novel alleles of *OsBADH2* through CRISPR/Cas9 mutagenesis and identify novel alleles of *OsBADH2* leading to production of aroma in a non-aromatic rice variety ASD16.

## Materials and methods

### Plant materials

A popular non-aromatic rice variety, ASD 16 was used in this study. Genetically pure seeds of ASD16 were obtained from Paddy Breeding Station, Tamil Nadu Agricultural University, Coimbatore, India and used in this study.

## Construction of CRISPR/Cas9 plant expression vectors

Nucleotide sequence encoding for functional *BADH2* (LOC_Os08g32870) was retrieved from MSU rice genome annotation project database (http://rice.plantbiology.msu.edu/). Synthetic guide RNA (*OsBADH2*-sgRNA:5'-TATGGCTTCAGCTGCTCCTA-3') targeting the upstream of previously reported 8-bp deletion on the 7[th] exon of *BADH2* [13] was designed using the E-CRISP tool (http://www.e-crisp.org). Oligomer of designed sgRNA was synthesized and cloned into pRGEB31 binary vector carrying Cas9 gene (Addgene plasmid # 51295; http://n2t.net/addgene:51295; RRID: Addgene_51295) [36]. BsaI was used to create the sticky ends in pRGEB31 vector and the sgRNA with BsaI hanging site was ligated. The ligated product was mobilized into XL1-blue strain and positive colonies were used for isolation of plasmid DNA.

## Rice transformation

The binary vector pRGEB31 harboring Cas9/*OsBADH2* sgRNA was mobilized into *Agrobacterium tumefaciens* LBA4404 by freeze-thaw method [37]. Immature embryos of rice variety ASD16 were collected from maturing panicles (15 days after anthesis) and used for *Agrobacterium* mediated transformation. After co-cultivation, putative transformed calli were kept in resting medium for a period of 5 days followed by the second round of resting for 10 days. Putative transformants were selected by subjecting the calli to two rounds of selection in a medium containing 50 mg/L hygromycin. Putative transgenic calli were regenerated in the presence of 1 mg/mL of NAA and 3 mg/mL 6-BA and rooted on half MS media containing 50 mg/L hygromycin [38]. Putative transgenic plants were transferred to transgenic green house at Tamil Nadu Agricultural University, Coimbatore, India (11.0152° N, 76.9326° E) for hardening and establishment.

## Molecular characterization of $T_0$ progenies

Genomic DNA was isolated from the leaves of all the $T_0$ progenies using CTAB method [39] and used for PCR analysis. All the lines were evaluated for the presence/absence of Cas9/sgRNA T-DNA through PCR using primers specific to hygromycin phosphotransferase (*Hpt*) gene and the *Cas9* gene(s). List of primers used in this study and the PCR profile followed for each primer combinations are listed in Table 1. PCR products were subjected to agarose gel electrophoresis for identifying putative edited plants based on amplicon size variations and subsequently by sequencing the PCR products. Sequencing data was analyzed using DSDecode [40], ICE v2 CRISPR analysis tool (https://ice.synthego.com/) and Clustal Omega [41]. Obtained results were confirmed by uploading the chromatogram files into TIDE analysis tool [42]. During TIDE analysis, default parameters were used and Indel range was set as 10. Proportion of edited DNA was calculated based on the sum of all significant Indels (P<0.001) detected by TIDE. Segregation of the Cas9/sgRNA T-DNA was investigated in $T_1$ generation.

## Screening of $T_0$ and $T_1$ progenies for aroma through sensory evaluation test

Putative gene (*OsBADH2*) edited plants of ASD16 were screened for the production of aroma through a sensory evaluation test using KOH-based method as described earlier [43]. About 10 cm long rice leaves from both non-transgenic and transgenic ASD16 were cut into small pieces and placed in petri dishes. Then 10 mL of 1.7% KOH solution was added onto the leaves. The petri dishes were covered and left for 20 to 30 minutes at room temperature and evaluated for the presence of aroma through sensory evaluation. Each sample was scored on a 1–4 scale, where 1 stands for absence of aroma, 2 for slight aroma, 3 for moderate aroma, and

**Table 1. List of primer combinations used for PCR analysis of recombinant bacterial colonies and putative genome edited rice plants.**

| S. No | Name of the Construct/organism | Name of the gene/ selection marker | Forward and Reverse primers | Annealing temperature | Size of the Amplicon |
|---|---|---|---|---|---|
| 1. | *Agrobacteriumtumefaciens* (LBA4404 strain) harboring pRGEB31-*BADH2*-sgRNA | *virG* | **virG-F:** 5'TCGATGTCGTGGTTCTTGAT3' | 56°C | 430 bp |
| | | | **virG-R:** 5' ATAAACCTCCTCGTCGCGTA 3' | | |
| | | *hpt* | **hpt–F:** 5' GCTGTTATGCGGCCATTGGTC 3' | 57.8°C | 686 bp |
| | | | **hpt–R:** 5' GCCTCCAGAAGAAGATGTTTG 3' | | |
| | | *CaMV35S* and *hpt* | **hpt–F:** 5' TACACAGCCATCGGTCCA 3' | 59°C | 1.3 kb |
| | | | **CaMV35S -R:** 5' ACCTCCTCGGATTCCATTGC 3' | | |
| 2. | *E.coli* harboring pRGEB31-*BADH2*-sgRNA | *M13* and *BADH2*-sgRNA | ***M13* -R:** 5'TCACACAGGAAACAGCTATG 3' | 55°C | 445 bp |
| | | | ***BADH2*-sgRNA-R:** 5'AAACTAGGAGCAGCTGAAGCCATA3' | | |
| | | *hpt* and *BADH2*-sgRNA | **hpt -F:** 5' TACACAGCCATCGGTCCA 3' | 55°C | 2,111 bp |
| | | | ***BADH2*-sgRNA-R:**5'AAACTAGGAGCAGCTGAAGCCATA3' | | |
| 3. | *Oryza sativa*(ASD16) | *CaMV35S* and *hpt* | ***CaMV35S*-R:** 5' ACCTCCTCGGATTCCATTGC 3' | 59°C | 1.3 kb |
| | | | **hpt–F:** 5' TACACAGCCATCGGTCCA 3' | | |
| | | GSP-*BADH2*-sgRNA | **GSP-*BADH2*-sgRNA F:** 5' TGCTCCTTTGTCATCACACCCT 3' | Step 3: 58°C | 390 bp |
| | | | **GSP-*BADH2*-sgRNAR:** 5'CCAAGTTCCAGTGAAACAGGCT3' | | |

4 for strong aroma. A panel of four persons smelled the samples and their evaluation were recorded.

## Profiling of volatile aromatic compounds in the seeds using GC-MS

Transgenic plants exhibiting strong aroma were selected and forwarded to $T_1$ generation. Presence of aroma was again confirmed through sensory evaluation test as described earlier. Seeds of two lines (# 8-19-2 and 8-19-6) producing strong aroma during sensory evaluation test were subjected to profiling of volatile compounds using GC-MS analysis along with seeds of non-transgenic ASD16. Grains were finely ground and added with 10 ml of 1.7% KOH solution. Elaborate protocol was described in our previous study [44]. In brief, the conical flask was tightly closed using a rubber cap containing a provision for a collector tube and sealed using the para film wrap to avoid leakage of volatiles. DB-5 ms capillary standard non-polar column with 30Mts dimension and Helium (He) as carrier gas were used. Sample collection tubes were removed from the flask and directly fed into the GC-MS. The analysis system is equipped with data acquisition and evaluation was carried out using Perkin Elemer Turbo Mass software Ver 6.1.0. Data analysis was carried out using Xcalibur™ (Thermo-Fisher Scientific) software. Obtained mass spectra was compared against NIST2014 (Version 2.2) library.

## Results

### Designing sgRNA targeting *OsBADH2*

For creating novel alleles of *OsBADH2* producing aroma in rice through CRISPR/Cas9 mutagenesis, sgRNA was designed near the 8 bp deletion on the exon 7 of *OsBADH2* using E-CRISP online tool (http://www.e-crisp.org). sgRNAs were synthesized with the adapter sequences specific to the forward primer (5'-GGCA-3') and reverse primer (5'-AAAC-3') as shown in Fig 1A. A schematic representation of the *Agrobacterium* T-DNA binary vector is shown in Fig 1B.

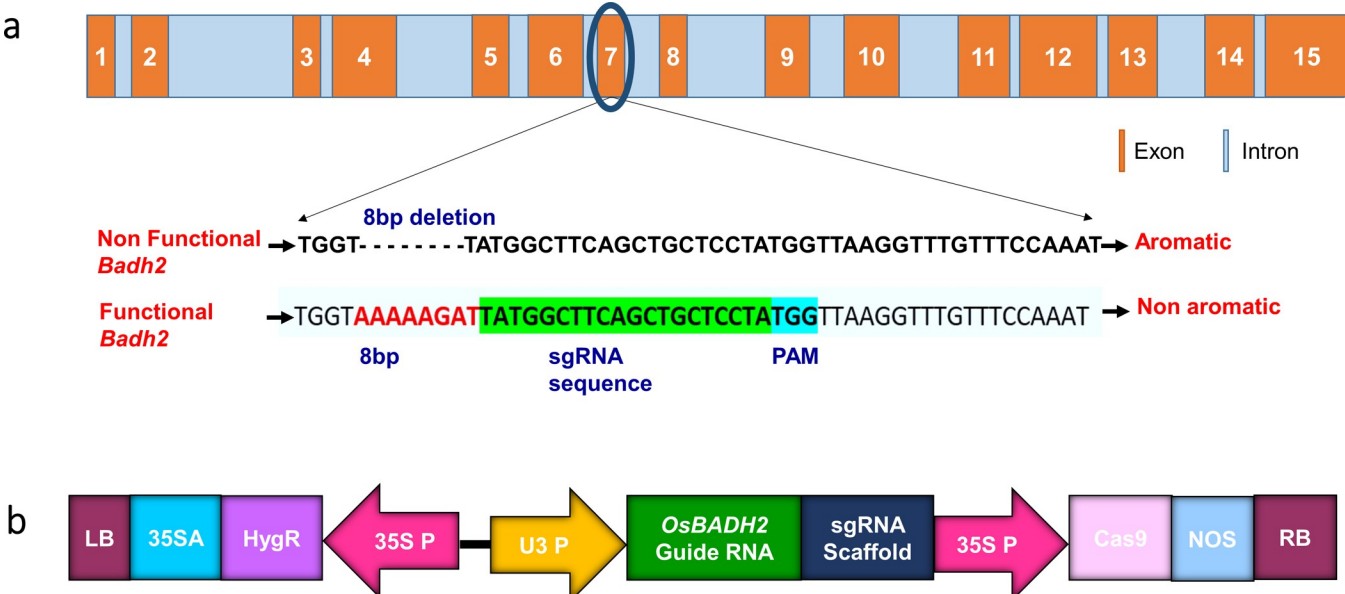

**Fig 1. Details of the target gene and vector used.** a) Structure of *OsBADH2* and target region on 7th exon chosen for designing sgRNA; b) Schematic representation of CRISPR/Cas9 vector pRGEB31 with sgRNA targeting *OsBADH2* used for stable *Agrobacterium*-mediated transformation of rice.

## Generation of gene edited plants of ASD 16

Immature embryos (15 DAF) were collected from a non-aromatic rice variety ASD16 and used for *Agrobacterium*-mediated transformation. About 400 immature embryos were used for co-cultivation using *Agrobacterium* harboring pRGEB31 binary vector engineered with *OsBADH2* sgRNA (S1 Fig). A total of 230 embryos were found to be healthy after two rounds of resting and subjected to selection. Around 120 calli were found to survive after two rounds of selection using hygromycin. The proliferating resistant calli were forwarded to pre-regeneration and then to regeneration medium (Fig 2A–2I). A total of 211 $T_0$ progenies belonging to 15 independent transgenic events were regenerated and transferred to rooting media. All the 211 $T_0$ progenies were screened for the presence of binary vector harboring Cas9-*OsBADH2* sgRNA and hygromycin expression cassette through PCR analysis using primers specific to the vector (reverse primer in the CaMV35 S and the forward primer in the *hpt* gene). All the 211 progenies were found to be positive for both vector and gene specific primers. In order to screen for the presence of mutations in the target loci, DNA isolated from all the 211 $T_0$ progenies were amplified using a set of primers flanking the sgRNA region on *OsBADH2*. Out of 211, ten progenies were found to possess insertions/deletions upon analysis through agarose gel electrophoresis. The amplicon size was ranging between 250–450 bp in the ten progenies whereas the non-transgenic showed an amplification of 390 bp (Figs 2J and S2).

## Sensory evaluation test identified $T_0$ progenies producing aroma

Sensory evaluation test was carried out in the leaf samples of all the 211 $T_0$ progenies using 1.7% KOH. Among the 211 $T_0$ progenies tested, thirteen lines belonging to six different events namely, Event# 1, 2, 7, 8, 9 and 12 were found to be aromatic (S3 Fig). Remaining 198 lines belonging to other nine events (Event # 3, 4, 5, 6, 10, 11, 13, 14 and 15) did not produce aroma in the $T_0$ generation.

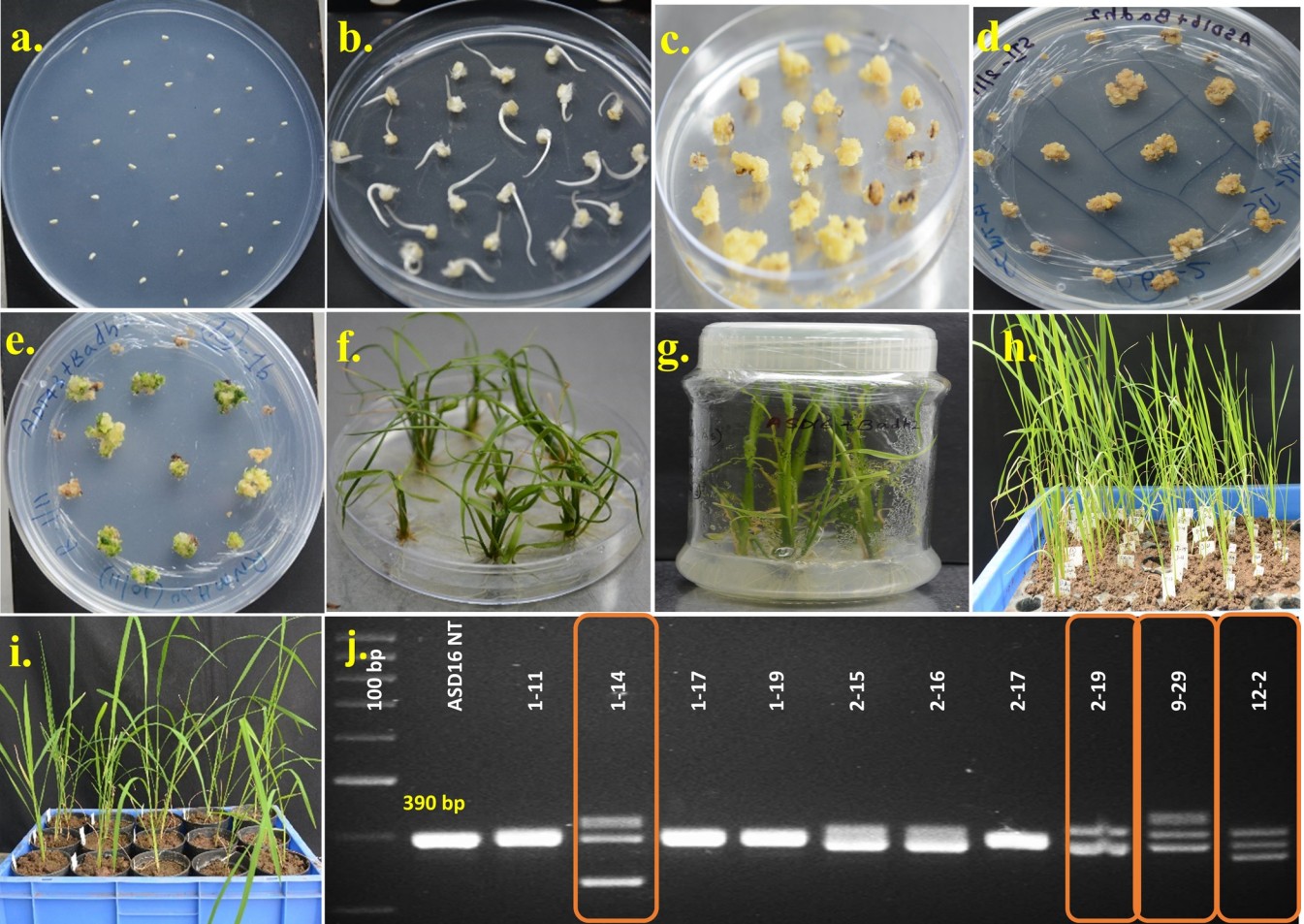

**Fig 2.** *Agrobacterium*-mediated transformation of immature embryos of rice cultivar ASD16 with pRGEB31 vector harbouring *BADH2*-sgRNA. a) Co-cultivated immature embryos; b) Immature embryo derived calli (7 days after co-cultivation); c) Resting stage; d) Selection using hygromycin; e) Regeneration of putative transformants; f) Shooting and root initiation; g) Second rooting; h) Hardening of putative transgenic plants in portrays; i) Establishment of putative transgenic plants under transgenic greenhouse conditions; j) Agarose gel electrophoresis showing presence of varying sizes of amplicons in the target region; the bars denotes the multi-allelic mutations.

## Molecular characterization of mutations in *OsBADH2* leading to aroma

All the thirteen progenies producing aroma in $T_0$ generation were subjected to PCR amplification using gene specific primers flanking the sgRNA and the amplicons were sequenced. Mutations were detected by comparing the sequences of edited lines against the non-transgenic ASD16 (NTASD16). Sequence analysis detected different types of mutations in the 13 aromatic lines as against the non-transgenic ASD16 (Fig 3). Among the thirteen progenies, eleven progenies showed bi-allelic mutations, one progeny showed multi-allelic and one progeny showed mono-allelic mutation (Table 2).

Sequencing results were further analyzed using ICE v2 CRISPR analysis tool to calculate the Indel percentage and knock-out score (Table 3). One of the edited lines, # 12–2 was found to possess a bi-allelic homozygous deletion mutation (-17/-17bp). Whereas, lines 1–11 (+1/-1), 1–17 (+1/-1), 1–19 (+1/-1), 2–15 (-5/-8), 2–16 (-8/-5), 2–17 (+1/-1), 2–19 (-8/-8), 7–1 (-4/-1), 7–2 (-4/-1), 8–19 (-2/-1), 8–27 (-2/-7), 9–29 (multi-allelic) and 11–7 (-1/+1) were found to harbor bi-allelic heterozygous mutations. Overall, all the 13 $T_0$ progenies producing aroma were

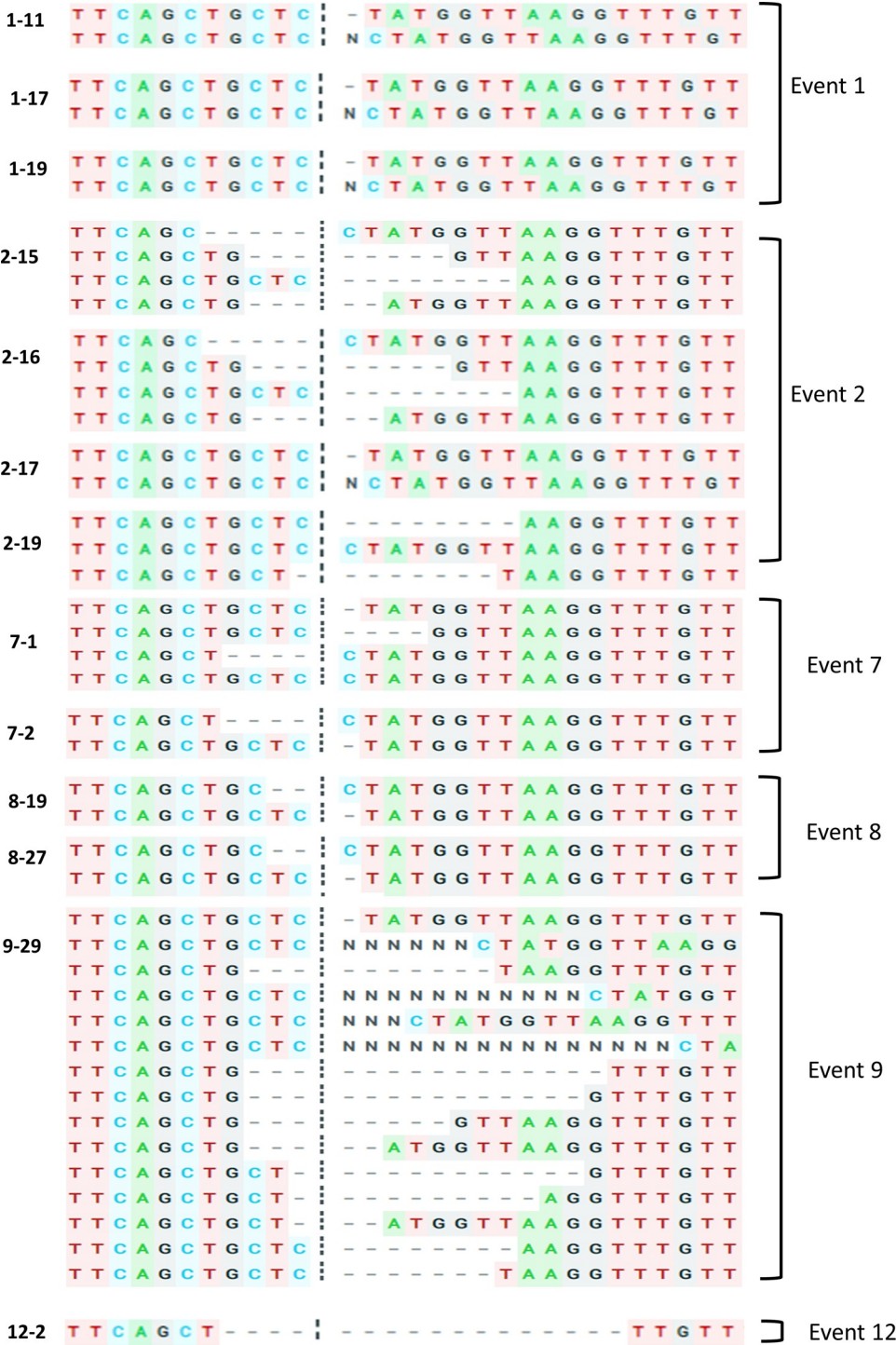

**Fig 3. Sequence analysis of target region in the T$_0$ mutant lines using ICE v2 CRISPR online tool.** Vertical black dots represent the predicted cleavage site of Cas9 and the horizontal black hyphen denotes deletions.

found to possess 22 different types of mutations in the vicinity of sgRNA (-17 bp to + 15 bp) region (Fig 4A and 4B). Out of the 22 mutations, -1/+1 bp mutation located 3 bp upstream from PAM region was found to be more frequent in five events (Event # 1, 2, 7, 8 and 9). This

**Table 2. Different types of mutations observed in T$_0$ progenies; Indels were predicted by using Synthego tool.**

| Sl.No | Line (T$_0$) | Mutant allele size (bp) | Mutation type |
|---|---|---|---|
| 1 | 1–11 | +1/-1 | Bi-allelic |
| 2 | 1–17 | +1/-1 | Bi-allelic |
| 3 | 1–19 | +1/-1 | Bi-allelic |
| 4 | 2–15 | -5/-8 | Bi-allelic |
| 5 | 2–16 | -5/-8 | Bi-allelic |
| 6 | 2–17 | +1/-1 | Bi-allelic |
| 7 | 2–19 | -8/-8 (at different position) | Bi-allelic |
| 8 | 7–1 | -1/-4 | Bi-allelic |
| 9 | 7–2 | -1/-4 | Bi-allelic |
| 10 | 8–19 | -2/-1 | Bi-allelic |
| 11 | 8–27 | -2/-1 | Bi-allelic |
| 12 | 9–29 | -1, +6,-10,+11,+3,+15, -15, -14, -8, -5, -12, -3,-8,-7 | Multi-allelic |
| 13 | 12–2 | -17 | Mono-allelic |

was followed by 8 bp deletion in the PAM region and 5 bp deletions at 1bp upstream from PAM region in two events (# 2 and 9). Tracking of Indels by Decomposition (TIDE) analysis confirmed the presence of targeted mutations in *OsBADH2* (S4 Fig). Predominant allele frequency was observed for all the events except line # 9 where multi-allelic mutation was observed. This might be due to chimeric events in the T$_0$ generation which require further analysis on T$_1$ and succeeding generations. R$^2$ value varied from 0.91–0.97 for heterozygous bi-allelic mutants and the total efficiency ranged between 56.6 to 96.8%.

**Table 3. Indel and Knock-out scores in the T$_0$ progenies predicted by using ICE v2 CRISPR analysis tool.**

| Sl.No | Sample | Indel (%) | Model Fit (R$^2$) | Knock-out Score |
|---|---|---|---|---|
| 1 | 1–11 | 98 | 0.98 | 98 |
| 2 | 1–17 | 98 | 0.98 | 98 |
| 3 | 1–19 | 98 | 0.98 | 98 |
| 4 | 2–15 | 97 | 0.97 | 97 |
| 5 | 2–16 | 100 | 1 | 100 |
| 6 | 2–17 | 98 | 0.98 | 98 |
| 7 | 2–19 | 63 | 0.96 | 63 |
| 8 | 8–19 | 98 | 0.98 | 98 |
| 9 | 12–2 | 100 | 1 | 100 |
| 10 | 7–1 | 87 | 0.92 | 87 |
| 11 | 7–2 | 89 | 0.89 | 89 |
| 12 | 8–27 | 98 | 0.98 | 98 |
| 13 | 9–29 | 69 | 0.69 | 45 |
| 14 | 11–7 | 97 | 0.97 | 97 |
| 15 | 2-16-1 | 96 | 0.96 | 96 |
| 16 | 2-16-2 | 97 | 0.97 | 97 |
| 17 | 2-16-3 | 96 | 0.96 | 96 |
| 18 | 8-19-1 | 100 | 1 | 100 |
| 19 | 8-19-2 | 98 | 0.98 | 98 |
| 20 | 8-19-6 | 98 | 0.98 | 98 |
| 21 | 11-7-3 | 97 | 0.97 | 97 |
| 22 | 11-7-7 | 100 | 1 | 100 |

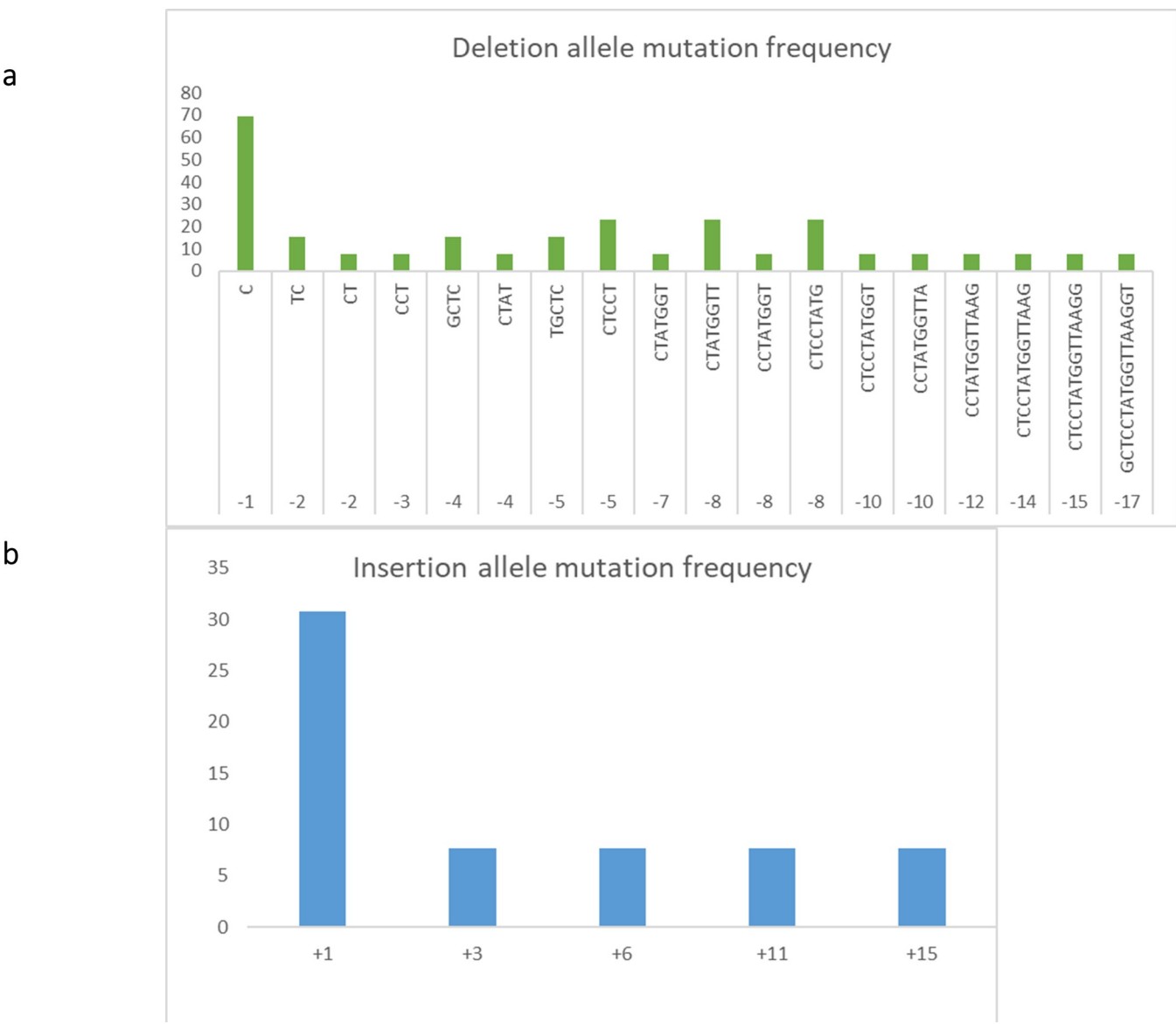

**Fig 4.** Frequency of deletion (a) and insertion (b) mutations observed among the thirteen $T_0$ lines.

### Inheritance of CRISPR/Cas9 induced mutations in the $T_1$ generation

To determine whether the CRISPR/Cas9 induced mutations in the *OsBADH2* and its influence on aroma were heritable, two events namely # 2–16 and # 8–19 which produced strong aroma in $T_0$ were forwarded to $T_1$ generation. Leaf sensory evaluation test in the progenies of two aromatic lines viz., # 8–19 (three plants namely 8-19-1, 8-19-2 and 8-19-6) and # 2–16 (three plants namely 2-16-1, 2-16-2 and 2-16-3) revealed the inheritance of aroma in all the 6 $T_1$ progenies. Similarly, these lines were subjected to Sanger sequencing and the mutant alleles were tracked for their inheritance. Among the three progenies of # 8–19 (harboring -2/-1 bp mutation in $T_0$), one line (# 8-19-1) was found to be homozygous for -2 bp mutation and the other two lines (8-19-2 and 8-19-6) were found to be still heterozygous for the bi-allelic (-2/-1 bp) mutation (Fig 5). In case of line # 2–16, all the three $T_1$ progenies (2-16-1, 2-16-2 and 2-

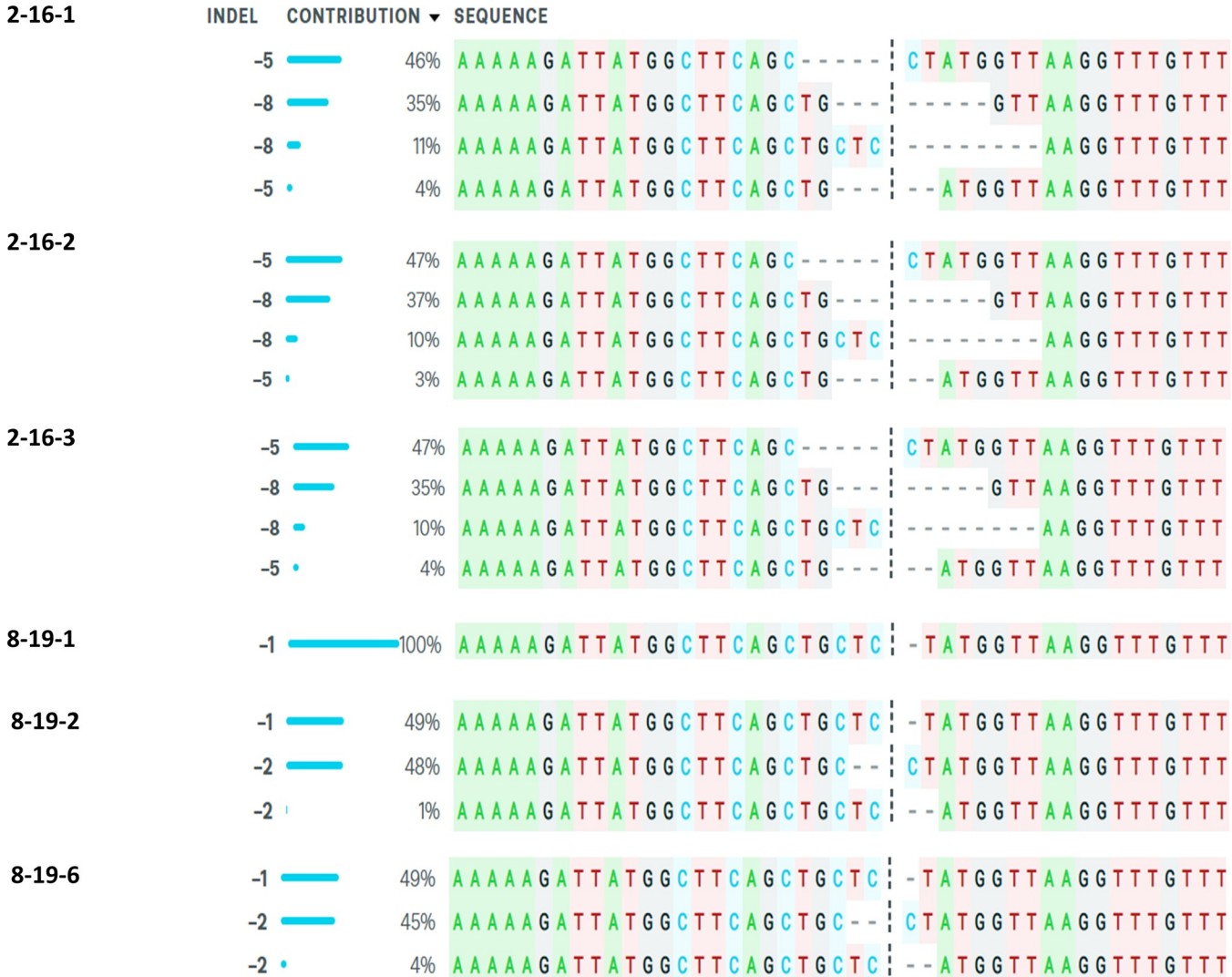

**Fig 5. Sequence analysis of target region using ICE v2 CRISPR tool in the $T_1$ generation identified 8-19-1 possessing homozygous mutation.** Vertical black dotted line represents the predicted cleavage site of Cas9 and the horizontal black hyphen denotes deletions.

16-3) were found to be still heterozygous for the bi-allelic mutation (-5/-8 bp). Indel percentage was found to be > 90% for homozygous mutations and 42.3 to 49.4% for the heterozygous mutations. The $R^2$ value ranged from 0.9 to 0.94 (S5 Fig).

## Comparative profiling of volatile compounds in the $T_1$ edited lines and non-transgenic ASD16

GC-MS analysis of volatiles released from seeds of two edited progenies # 8-19-2 and 8-19-6 identified accumulation of several novel aromatic compounds in the edited progenies (S1 Table). Important aromatic compounds namely, pyrrolidine, pyridine, pyrazine, pyradazine and pyrozole were present in the grains of edited progenies (S1 Table). Analysis of retention time and area percentage of peaks pertaining to the novel aromatic compounds clearly indicated that these compounds were not present in the non-transgenic ASD16 where as they were present at detectable limits in the edited lines producing aroma (Fig 6). Among the novel

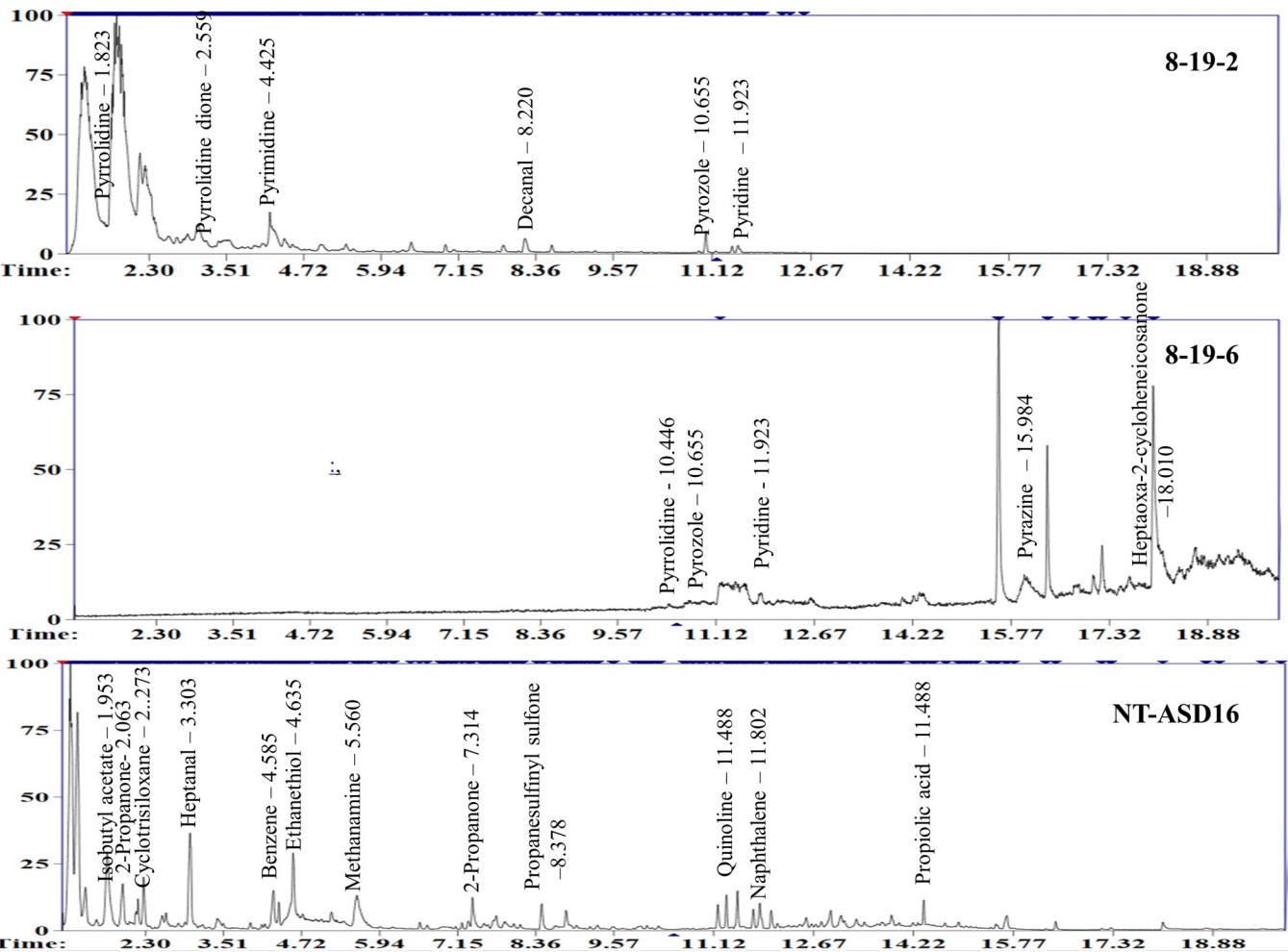

**Fig 6. Chromatogram of aromatic compounds detected in the extracts of non-transgenic ASD16 and two edited lines viz., # 8-19-2 and 8-19-6.**

aromatic compounds detected, pyrrolidine is the precursor molecule in the synthesis of 2AP and pyridine shares structural homology with 2AP. Apart from the above compounds, aromatic compounds such as nonanal, octanal, hexanal, propanal, dodecane, undecane, tetradecane, butanone, pyrrole, dioxolane, propanone, octenal, acetophenone, propenal, styrene and decanal were also found only in the grains of edited lines.

## Discussion

### CRISPR/Cas9 induced novel alleles of *OsBADH2*

Physical and cooking qualities of rice grains determine the marketability and acceptability of rice varieties. Aroma is one of the key traits determining the price of rice in both local and international market. Apart from Basmati genotypes possessing long slender grains, only few other medium/short slender grain rice varieties possessing aroma are in the market. Those short/medium slender aromatic genotypes are not high yielding and possess several other disadvantages. Development of aromatic rice varieties possessing superior grain qualities through conventional or molecular breeding approaches takes considerable number of years and in some cases retaining the superior grain qualities of elite genotypes still remains a challenge. In

this context, creating mutations leading to aroma in high yielding elite genotypes through CRISPR/Cas9 seems to be an attractive strategy to overcome the number of years required for developing desired genotypes and also to overcome the problems due to linkage drag [45, 46].

Several attempts have been made to unravel biochemical, genetic and molecular basis of aroma in rice which led to the identification of a major volatile compound 2-acetyl 1-pyrroline responsible for aroma in rice [13, 47–50]. Lorieux *et al.* [9] identified one major QTL on chromosome 8 and two minor QTLs (one on chromosome 4 and another on chromosome 12) linked to accumulation of 2AP. Similarly, Amarawathi *et al.* [17] identified 3 loci, one each on chromosome 3 (*ARO3.1* with 6.1% PVE), chromosome 4 (*ARO4.1* with10.3% PVE) and chromosome 8 (*ARO8.1* with 18.9% PVE). Subsequent studies led to the identification of a single recessive gene encoding betaine aldehyde dehydrogenase 2 (*BADH2*) on chromosome 8 to be responsible for aroma in scented rice [13]. Majority of the aromatic rice genotypes possess an 8 bp deletion in exon 7 of *badh2* gene [51]. Based on this, a breeder friendly Indel marker discriminating aromatic and non-aromatic rice genotypes was developed [17] which enabled accelerated development of aromatic rice genotypes through MAS programs. Shao *et al.* [52] made a thorough survey of allelic variants of *OsBADH2* in a set of 144 aromatic rice varieties and found 6 different haplotypes of *OsBADH2* viz., i). *BADH2.2* (7 bp deletion on exon 2); ii) *BADH2.2* (75 bp deletion on exon2); iii) *BADH2.4–5* (806 bp deletion on exon4 and exon5); iv) *BADH2.7* (8 bp deletion on exon 7); v) *BADH2.10* (G to A on exon 10) and vi) *BADH2.13* (C to T on exon 13). Interestingly all the 6 different haplotypes were producing aroma. This indicated that different types of mutations in *OsBADH2* can lead to accumulation of aroma in rice. Creating desired mutant phenotype through conventional breeding is a challenging task which will involve generation and screening of several thousands of mutant progenies. Genome editing is a controlled site-specific process that allows editing of DNA sequences using molecular tools viz., Zinc finger nucleases (ZFNs), Transcription Activator Like effector nucleases (TALENs) and Clustered Regulatory Interspaced Short Palindromic Repeats (CRISPR) associated (CRISPR/Cas), CRISPR/Cas is preferred over other site specific nucleases due to its simplicity, flexibility and accuracy [53].

During recent years, several research papers have reported the diverse applications of genome editing using the CRISPPR/Cas9 [54]. Genome editing for multiple susceptible genes have been targeted to confer broad spectrum resistance against blight disease in rice [55–57]. Subsequently, CRISPR/Cas9 was applied for improving important quantitative and qualitative traits in rice including editing of *OsSAPK2* regulating drought responses [58], *OsGIF1* regulating size of stem, leaves and grain [59], *OsMGD2* regulating grain quality [60], *OsARM1* controlling tolerance against for arsenic toxicity [61], *OsPPa6* contributing for tolerance against alkalinity [62] and *OsBADH2* leading to aroma in rice [35]. Based on the above evidences, this study was undertaken with an aim of demonstrating the efficacy of CRISPR/Cas9 in accelerated development of aromatic rice genotypes and secondly to create different mutations in *OsBADH2* leading to accumulation of aroma in rice.

In this study, CRISPR/Cas9 tool was used to create mutations in *OsBADH2* of a non-aromatic rice variety ASD16. Rice genome harbors two homologs of BADH namely, BADH1 (LOC_Os04g39020) and BADH2 (LOC_Os08g32870) which share 75.94% sequence similarity [64]. BADH1 was reported to be involved in modulating abiotic stress responses and BADH2 was found to be involved in the production of aroma [63, 64]. sgRNA targeting BADH2 was designed carefully after comparing the sequences of both the homologs and the designed sgRNA targeting BADH2 shared less than 50% similarity with BADH1 (S6 Fig). PCR analysis of 211 $T_0$ progenies confirmed the presence of vector backbone pRGEB31 in all the putative edited lines. Presence of amplicons in 10 putative edited lines having molecular weight deviating from the expected size (390 bp) indicated the presence of mutations in those lines (Figs S2

and 2J). Sensory evaluation test conducted among the 211 putative gene edited $T_0$ progenies identified 13 progenies producing various degrees of aroma (S3 Fig). Cloning and sequencing of target regions of all the 13 aromatic progenies identified 22 different kinds of mutations (Fig 4). Among the various kinds of mutations, -1/-2 bp deletions present in the line # 8–19 and -8 /-5 bp deletion present in the line # 2–16 were found to produce strong aroma and thus indicating significant level of 2AP accumulation. Screening of $T_1$ progenies revealed the stable inheritance of aroma in the progenies of above two lines. Among the 13 $T_0$ progenies producing aroma, the reported mutation (8 bp deletion in the exon 7) was not present in any of the lines. Thus, all the 13 could be novel alleles of fragrance gene (*OsBADH2*) in rice and can be utilized in breeding programs. Studies have reported the presence of alleles other than the traditionally reported 8bp deletion of *OsBADH2*. Further, valedictory experiments are needed for confirming the association of 13 novel mutations with aroma in rice.

## Mutations in *OsBADH2* led to the alteration in the aroma profile

Several studies have documented the profile of volatile compounds in aromatic rice varieties. However, 2AP is reported to be the main volatile compound responsible for the production of aroma in rice [5]. In the present study, several allelic variants of *OsBADH2* associated with aroma have been identified. One of the mutations (-1/-2 bp deletion) present in # 8–19 produced strong aroma in both $T_0$ and $T_1$ generation. GC-MS analysis in the seeds of aromatic progenies of ASD16 (# 8-19-2 and # 8-19-6) showed the presence of altered profile of aromatic compounds when compared to non-transgenic ASD16. The comparative profiling of volatile compounds in the seeds of $T_1$ progenies showed the presence of several unique aromatic compounds viz., pyrrolidine, pyridine, pyrazine, pyradazine, pyrozole, pyrrolidine methanol and 2,5-pyrrolidinedione [5]. Pyrrolidine is a direct precursor of 2AP [65].

Pleasant aroma in the grains of basmati and other short grain aromatic rice genotypes is due to the blend of more than 500 volatile aromatic compounds [66]. In this study also, several aromatic compounds belonging to the class of aldehydes, ketones, organic acids, esters, alcohols, aromatic hydrocarbons, terpenes, alkenes, pyridines and N-heterocyclic compounds were detected only in the seeds of gene edited and aromatic $T_1$ progenies. These chemical compounds were used to distinguish between scented and non-scented rice cultivars [5]. Other heterocyclic aromatic compounds such as pyridine, pyrazole, pyrazoline, pyrimidinone, pyrrolidinedione and furan found in the seeds of aromatic progenies share a common 5' carbon ring similar to 2AP.

Similarly, other additional volatile compounds found in our current study were reported in aromatic rice varieties. For instance, propanal possesses sharp musty odor like rubbing alcohol [67] and octanal is a fragrant liquid with a fruit-like odor, it occurs naturally in citrus oils. It is used commercially as a component in perfumes and in flavor production for the food industry [68]. Whereas, nonanal possess rose-orange odor (https://pubchem.ncbi.nlm.nih.gov/compound/Nonanal), decanal is an odor agent smells similar to that of orange peel and acetophenone was present in 8-19-6 which possess sweet pungent taste and odor resembling the odor of oranges (https://pubchem.ncbi.nlm.nih.gov/compound/Decanal).

In summary, CRISPR/Cas9 system has been successfully applied to create new allelic variations *OsBADH2* gene. The allelic variations we observed in the exon 7 of *BADH2* in the current study may be contributing to aroma in the non-aromatic rice variety ASD16 and further functional analysis along with phenotyping may confirm the hypothesis. Studies have reported that the fragrance of cooked rice consists of more than 200 volatile compounds such as hydrocarbons, alcohols, aldehydes, ketones, acids, esters, phenols, pyridines, pyrazines, and other compounds [69]. Therefore, along with the traditional volatile compounds other aromatic

compound we observed might be the effect of new alleles created in this study. However further analysis on all the $T_1$ lines and succeeding generations may reveal the role of the newly created novel alleles for aroma in rice. The heterozygous mutants obtained from the current study will be useful to obtain homozygous mutants in the next generation and characterization of the new alleles for aroma trait will pave way to improve aroma in agronomically important non-aromatic rice varieties.

## Conclusion

This study has demonstrated the efficiency of CRISPR/Cas9 in creating novel alleles of BADH2 leading to aroma in rice. Conventional/MAS breeding strategies require at least half a decade of time to develop aromatic rice varieties. Utilization of CRISPR/Cas9 tool has shortened the time required in developing aromatic ASD16 lines. Developed lines will serve as novel genetic stocks/donors in breeding programs for developing non-basmati aromatic rice varieties. Further experiments will focus on characterization of aroma profiles of gene edited lines of ASD16 thereby to identify novel aromatic compounds in rice and to understand the molecular basis of aroma production in rice.

## Supporting information

**S1 Fig. Details of pRGEB31 harbouring *BADH2*-sgRNA under *OsU3* promoter.** a) pRGEB31 possessing *OsBadh2* guide RNA b) Sequence analysis showing the presence of *OsBADH2*-sgRNA in pRGEB31.
(TIF)

**S2 Fig. Agarose gel electrophoresis showing amplification pattern of *OsBADH2* fragments in the $T_0$ progenies.**
(TIF)

**S3 Fig. Sensory evaluation test for assessing the production of aroma in the selected $T_0$ lines.**
(TIF)

**S4 Fig. Screening of $T_0$ progenies for targeted mutations.** Prediction of putative bi-allelic mutants using TIDE analysis.
(TIF)

**S5 Fig. Inheritance percentage of Indels in the $T_1$ progenies using TIDE software.**
(TIF)

**S6 Fig. Multiple sequence alignment of *BADH1* (LOC_Os04g39020.1), functional BADH2 (LOC_Os08g32870.1 (F) and non-functional *BADH2* (LOC_Os08g32870.1 (NF)).**
(TIF)

**S1 Table. List of volatile compounds identified in seeds of non-transgenic ASD16 and gene edited lines.**
(XLSX)

**S1 Raw images.**
(PDF)

## Author Contributions

**Conceptualization:** Raveendran Muthurajan.

**Data curation:** Valarmathi Ramanathan, Rohit Kambale.

**Formal analysis:** Shanthinie Ashokkumar.

**Methodology:** Shanthinie Ashokkumar, Valarmathi Ramanathan, Hifzur Rahman, Raveen-dran Muthurajan.

**Software:** Deepa Jaganathan, Rakshana Palaniswamy.

**Supervision:** Raveendran Muthurajan.

**Validation:** Hifzur Rahman, Rakshana Palaniswamy.

**Writing – original draft:** Deepa Jaganathan, Rohit Kambale.

**Writing – review & editing:** Raveendran Muthurajan.

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
