## [Decision Letter · Decision Letter 0]

16 Jun 2020

PONE-D-20-13769

Creation of novel alleles of fragrance gene OsBADH2 in rice through CRISPR/Cas9 mediated gene editing

PLOS ONE

Dear Dr. Muthurajan,

Thank you for submitting your manuscript to PLOS ONE. After careful consideration, we feel that it has merit but does not fully meet PLOS ONE’s publication criteria as it currently stands. Therefore, we invite you to submit a revised version of the manuscript that addresses the points raised during the review process.

ACADEMIC EDITOR: The manuscript entitled “Creation of novel alleles of fragrance gene OsBADH2 in rice through CRISPR/Cas9 mediated gene editing” is well written and recommended for publication with minor revision. The suggested edits and comments addressed by the reviewers need to be addressed by the authors in the updated version. The introduction part needs to be strengthened with additional details on aroma history with relevant recent citations and references. If possible the effects on growth vigor, stress response in rice edited sequenced rice progenies are to be addressed for clear validation and to strengthen the quality of the manuscript. Clear English rephrasing throughout the manuscript will improve the overall presentation of the manuscript.

We look forward to receiving your revised manuscript.

Kind regards,

Ramasamy Perumal, Ph.D.

Academic Editor

PLOS ONE

Reviewers' comments:

Reviewer's Responses to Questions

**Comments to the Author**

1. Is the manuscript technically sound, and do the data support the conclusions?

Reviewer #1: Yes

Reviewer #2: Yes

2. Has the statistical analysis been performed appropriately and rigorously? 

Reviewer #1: Yes

Reviewer #2: Yes

3. Have the authors made all data underlying the findings in their manuscript fully available?

Reviewer #1: Yes

Reviewer #2: Yes

4. Is the manuscript presented in an intelligible fashion and written in standard English?

Reviewer #1: Yes

Reviewer #2: Yes

5. Review Comments to the Author

Reviewer #1: The manuscript-Creation of novel alleles of fragrance gene OsBADH2 in rice through CRISPR/Cas9 mediated gene editing by Ashokkumar et al., is well organized and reports the exploitation of genome-editing tool to develop value-added economically important traits in rice.

I have the below mentioned a few major concerns.

The authors should include the aroma chemistry in the introduction to explain the function of BADH2 protein in detail, and how it may regulate the aroma threshold in rice.

I was wondering if the authors have observed any other effects in rice progenies with edited BADH2? Growth vigor, stress response, etc.

The authors did not mention other BADH's in rice. Two BADH homologs are present in rice, BADH1, and BADH2, with ~75% sequence similarity between them. BADH1 is associated primarily with abiotic stress tolerance, while BADH2 is known to attribute to rice aroma. The authors should mention such critical aspects related to BADH.

The authors performed comparative profiling of volatile compounds in the T1 edited lines; this is the critical result in context with the study objective. The authors must present the change in the pyrrolidone levels in edited lines compared with the non-transgenic ASD16 line. These results should be in the main text.

The conclusion is very superficial. The authors can improve the conclusion section more previously.

In the second paragraph, aro8.1 and badh2 should be in uppercase, also several other places the gene names mentioned in the lowercase; please check.

Reviewer #2: Grain quality is an important trait which can improved through genome editing and the results presented in the paper seems to be convincing. However certain points in the manuscript needs to be addressed

1. Initial PCR screening shows altered allelic pattern in 10 progenies, while 13 progenies were sequenced to check the mutation. On what basis 13 progenies were selected for sequencing.

2. The supplementary fig s2 indicates multiple bands in many progenies, why those lines were not highlighted

3. Justify the variation existing within the progenies of same event for ex. 2-17/2-19, 7-1/7-2

4. Only two lines were forwarded and screened in the T1 generation, whether other progenies with mild aroma were tested in T1 generation. The probability of generating a vector free mutant line can be increased through screening several progenies.

5. The frequency of +1/-1 mutation seems to be high, which does not lead to strong phenotype.

6. English language correct will improve the MS

6. PLOS authors have the option to publish the peer review history of their article (what does this mean?). If published, this will include your full peer review and any attached files.

Reviewer #1: Yes: Umesh K Reddy

Reviewer #2: No

---

## [Author Response · Author response to Decision Letter 0]

14 Jul 2020

We have attached a separate letter indicating the responses to the reviewers

---

## [Editor Report · Decision Letter 1]

20 Jul 2020

Creation of novel alleles of fragrance gene OsBADH2 in rice through CRISPR/Cas9 mediated gene editing

PONE-D-20-13769R1

Dear Dr. Raveendran Muthurajan,

We’re pleased to inform you that your manuscript has been judged scientifically suitable for publication and will be formally accepted for publication once it meets all outstanding technical requirements.

Kind regards,

Ramasamy Perumal, Ph.D.

Academic Editor

PLOS ONE

Additional Editor Comments (optional):

Authors addressed all the comments by the reviewers in the revised version. The manuscript is accepted recommended for publication as full research article in PLOS ONE.
---

## [Editor Report · Acceptance letter]

23 Jul 2020

PONE-D-20-13769R1 

Creation of novel alleles of fragrance gene OsBADH2 in rice through CRISPR/Cas9 mediated gene editing 

Dear Dr. Muthurajan:

I'm pleased to inform you that your manuscript has been deemed suitable for publication in PLOS ONE. Congratulations! Your manuscript is now with our production department. 

Kind regards, 

on behalf of

Dr. Ramasamy Perumal 

Academic Editor

PLOS ONE